# A Novel Staggered Double-Segmented Grating Slow-Wave Structure for 340 GHz Traveling-Wave Tube

**DOI:** 10.3390/s23104762

**Published:** 2023-05-15

**Authors:** Zechuan Wang, Junwan Zhu, Zhigang Lu, Jingrui Duan, Haifeng Chen, Shaomeng Wang, Zhanliang Wang, Huarong Gong, Yubin Gong

**Affiliations:** 1National Key Laboratory of Science and Technology on Vacuum Electronics, School of Electronic Science and Engineering, University of Electronic Science and Technology of China, No. 2006 Xiyuan Avenue, High-Tech District (West District), Chengdu 611731, China; 202122022214@std.uestc.edu.cn (Z.W.); zhujunwan@yeah.net (J.Z.);; 2Yangtze Delta Region Institute (Huzhou), University of Electronic Science and Technology of China, Huzhou 313001, China

**Keywords:** traveling-wave tube, slow-wave structure, staggered double-segmented grating, high interaction impedance, low ohmic loss

## Abstract

In this paper, a novel staggered double-segmented grating slow-wave structure (SDSG-SWS) is developed for wide-band high-power submillimeter wave traveling-wave tubes (TWTs). The SDSG-SWS can be considered as a combination of the sine waveguide (SW) SWS and the staggered double-grating (SDG) SWS; that is, it is obtained by introducing the rectangular geometric ridges of the SDG-SWS into the SW-SWS. Thus, the SDSG-SWS has the advantages of the wide operating band, high interaction impedance, low ohmic loss, low reflection, and ease of fabrication. The analysis for high-frequency characteristics shows that, compared with the SW-SWS, the SDSG-SWS has higher interaction impedance when their dispersions are at the same level, while the ohmic loss for the two SWSs remains basically unchanged. Furthermore, the calculation results of beam–wave interaction show that the output power is above 16.4 W for the TWT using the SDSG-SWS in the range of 316 GHz–405 GHz with a maximum power of 32.8 W occurring at 340 GHz, whose corresponding maximum electron efficiency is 2.84%, when the operating voltage is 19.2 kV and the current is 60 mA.

## 1. Introduction

As a popular research topic in the field of electromagnetic wave science, benefiting from its superiority in permeability, controllability, and transmissibility, the terahertz wave is widely utilized in communication systems, imaging fields, and biomedical fields [1]. In the field of terahertz science, how to generate the terahertz wave is a very key issue [2,3,4]. As highly effective broadband high-power signal sources, vacuum electronic devices (VEDs) are widely used. As one of many VEDs, the traveling-wave tube (TWT) is widely used as a broadband high-power amplifier. As the main site of interaction between an electromagnetic wave and an electron beam, the slow-wave structure (SWS) has a large impact on the performance of the TWT [5,6].

Currently, the main SWSs used for TWTs at 340 GHz include the folded waveguide (FW) [7,8,9], staggered double-grating (SDG) [10,11,12], sine waveguide (SW) [13,14,15], and deformations of the above three SWSs. However, with the reduction in SWS size caused by the increase in operating frequency, the sizes of electron beam tunnels become progressively smaller, which will limit the improvement of beam current and output power; at the same time, the ohmic loss of the metal also increases, due to skin depth and fabrication accuracy. In order to reduce the impact of these two problems, on the one hand, a sheet electron beam [16,17,18] that has a larger dimension should be considered for interaction with the SWS to reduce the impact of the reduced size of the SWS, and, on the other hand, a new SWS with low transmission loss should be chosen as the site of beam–wave interaction.

Due to the natural sheet beam tunnel, low ohmic loss, and wide operating band, the SW has become a research hotspot in recent years [13,15]. Thus, SW is a suitable SWS for submillimeter wave TWTs [19]. However, the relatively low interaction impedance of the SW-SWS will affect the improvement of SW-TWT performance in terms of output power, gain, and electron efficiency. Therefore, improving the interaction impedance of the SW-SWS while retaining the advantages of a wide operating band and low ohmic loss is a worthwhile research issue [14,19,20].

The SW-SWS, as a modification of the SDG-SWS, has the characteristics of low reflection and low ohmic loss, compared with the SDG-SWS, but its interaction impedance is lower than that of the SDG-SWS. Therefore, based on the comprehensive analysis of the SW-SWS, with low ohmic loss, and the SDG-SWS, with high interaction impedance, a novel staggered double-segmented grating (SDSG) SWS is proposed by innovatively introducing the rectangular geometric ridges of the SDG-SWS into the SW-SWS. The new SDSG-SWS combines the advantages of the above two SWSs and maintains the characteristics of lower ohmic loss and higher interaction impedance.

The remainder of the article is arranged as follows: The high-frequency electromagnetic characteristics of the SDSG-SWS are analyzed in Section 2; Section 3 describes the particle-in-cell (PIC) simulation of the beam–wave interaction of TWT using the SDGS-SWS and analyzes the results; in Section 4, the related conclusions are drawn.

## 2. Design and Analysis

For the SWSs, the capacity of the electromagnetic wave to exchange energy with the electron beam is generally characterized by the interaction impedance $K_{c}$, which is defined as:(1)$$K_{c}=\frac{E_{zn}E_{zn}^{*}}{2\beta_{n}^{2}v_{g}U}$$

Here, $E_{zn}$ is the longitudinal electric field component of the *n*th spatial harmonic, $E_{zn}^{*}$ is the conjugate value for $E_{zn}$, U is the system energy storage per unit length, $v_{g}$ is the group velocity of electromagnetic wave transmission, and the propagation constant for the *n*th spatial harmonic is $\beta_{n}$.

For the TWTs, obtaining a greater output power over a wide operating frequency band is required under the same operating conditions. Therefore, the dispersion of the SWS is designed at the same level for the comparison of TWTs with different SWSs; that is, the transmission characteristics for the different SWSs are the same. Of course, the synchronous voltage, also known as the operating voltage, is the same. Subsequently, the research on the electric field distribution of SWSs is a key focus, which is closely related to interaction impedance and ohmic loss. According to Formula (1), and based on the previous analysis, for the different SWSs, the denominator of (1), which is dominated by the dispersion, is almost the same, and improving the interaction impedance depends entirely on the electric field distribution of SWSs. 

Figure 1 shows the distribution of the longitudinal electric field $E_{Z}$ for the SW-SWS and SDG-SWS, respectively. As observed in Figure 1a, the $E_{Z}$ is mainly concentrated at the bend of the SW-SWS, which is comparable to that of the SDG-SWS in Figure 1b. However, the $E_{Z}$ in region I of the SDG-SWS is significantly stronger than that in the same region of the SW-SWS, which is due to the SDG-SWS having rectangular geometric ridges. Thus, a new idea was proposed: to introduce the rectangular geometric ridges of the SDG-SWS into the SW-SWS in order to improve the interaction impedance of the SW-SWS. Based on the above idea, the SDSG-SWS is proposed.

Figure 2a,b present the three-dimensional solid models (removing the side wall at *X*-max) of the SDSG-SWS and SW-SWS, respectively, while Figure 2c,d show the left view and sectional view, respectively, in the *y-o-z* plane of the SDSG-SWS with the sheet electron beam (the red part is the sheet electron beam). As observed in Figure 2, both the SDSG-SWS and SW-SWS have equal cross-sectional features (b×w), and the SDSG-SWS has the same rectangular geometric ridges as the SDG-SWS at the top and bottom of the metal grating. To better illustrate the origin of the SDSG-SWS, the detailed evolution from SW-SWS to SDSG-SWS is presented in Figure 3.

The transition structure presented in Figure 3b with the same equal cross-section characteristics as the SW-SWS can be obtained by introducing the rectangular geometric ridges of the SDG-SWS at the top and bottom of the sinusoidal-shaped grating of the SW-SWS presented in Figure 3a. Considering the manufacturing accuracy and processing difficulty of the transition structure in the submillimeter wave band, the sinusoidal profile of the grating in Figure 3b is replaced by a linear profile, and the SDSG-SWS presented in Figure 3c is obtained. The SDSG-SWS is the combination of the SW-SWS and SDG-SWS. The SDSG-SWS is obtained by introducing the rectangular geometric ridges of the SDG-SWS while keeping the cross-sectional characteristics of the SW-SWS. Therefore, it can be predicted that the SDSG-SWS should have almost the same ohmic loss and dispersion characteristics as the SW-SWS, but its interaction impedance should be higher than that of the SW-SWS.

In order to verify the above speculation and illustrate the advantages of the SDSG-SWS over the SW-SWS, the normalized phase velocities of both SWSs should be kept at the same level within the same frequency band, which is a prerequisite for the comparison. The optimized parameters are presented in Table 1. The electromagnetic characteristics of the two SWSs are calculated using the 3D simulation software Ansoft High Frequency Structure Simulator. The dispersion, attenuation constant, and interaction impedance calculated are presented in Figure 4, Figure 5 and Figure 6.

Figure 4 shows the dispersion curves for the SDSG-SWS and SW-SWS. The results show that, in a fairly wide frequency range, the normalized phase velocities are essentially the same when their parameters are optimized. Based on these, Figure 5 shows the attenuation constants of the SDSG-SWS and SW-SWS, and the results show that the attenuation constants of both SWSs are also essentially equal for the same dispersion. The ohmic loss of SWSs is represented by the attenuation constant. The results prove that the introduction of rectangular geometric ridges does not change the low ohmic loss characteristics of the SW-SWS under the same dispersion.

Comparison of interaction impedance for both SWSs is presented in Figure 6. The results show that the SDSG-SWS has a higher interaction impedance compared with the SW-SWS. In the band range of 310 GHz–420 GHz, the minimum value of interaction impedance is 0.57 Ohm for the SDSG-SWS and 0.34 Ohm for the SW-SWS, an improvement of 59.6%. The improved interaction impedance means that the electric field can better exchange energy with the electron beam, which can effectively improve the power, gain, and electron efficiency of the TWT.

To further illustrate that the enhancement of the $E_{Z}$ is the reason for the enhancement of the SDSG-SWS’s interaction impedance, by using the CST eigenmode solver, the $E_{Z}$ along the black lines shown in Figure 7 (A–B and C–D) is calculated, and the results are shown in Figure 8. It can be observed that, for the SW-SWS, the $E_{Z}$ is strongest near the bend and gradually decreases as the distance from the bend increases; the trend of the $E_{Z}$ for the SDSG-SWS is comparable to that for the SW-SWS. However, the calculated value of the $E_{Z}$ is higher than that of the SW-SWS from Figure 8. The results indicate that the SDSG-SWS has a larger longitudinal electric field $E_{Z}$ at the place of its electron beam tunnel. 

The vacuum model of the SDSG-SWS with couplers is shown in Figure 9a, which is mainly composed of a beam tunnel, the main slow-wave circuit, the mode converter, and the input–output waveguide. The main slow-wave circuit consists of 120 cycles. The operating mode of the SDSG-SWS is generally the EH mode, while the mode of input–output waveguide is $TE_{10}$ mode. Therefore, the mode converter is designed to convert the $TE_{10}$ mode to the EH mode in order to ensure that the input signal can be effectively coupled into the slow-wave circuit and stably amplified without reflection. In Figure 9b, it can be observed that the length of the mode converter is four periods, in which the height of gratings decreases proportionally towards the centerline of gratings until it becomes a smooth rectangular waveguide. As shown in Figure 9b, the electric field can gradually change from EH mode to $TE_{10}$ mode through the coupler.

According to the model shown in Figure 9a, the calculation results of electromagnetic transmission characteristics of the SDSG slow-wave circuit are shown in Figure 10. From 319 GHz to 438 GHz, $S_{11}$ is below −17.9 dB, while $S_{21}$ is above −15 dB.

## 3. Beam–Wave Interaction Simulation

The performance of the SDSG-TWT and the SW-TWT were analyzed using the PIC simulation of CST Particle Studio. In the PIC simulation, oxygen-free copper was used as the circuit material, and its conductivity is $1.8\times 10^{7}\;S/m$, considering the distribution loss of the circuit. In order to illustrate the advantages of TWTs using the SDSG-SWS in saturated power, gain, and electron efficiency, TWTs using the SDSG-SWS and SW-SWS should be kept at the same operating voltage and current. According to the dispersion characteristics shown in Figure 3, the synchronous operating voltage of both TWTs is set to 19.2 kV, and the operating current is set to 60 mA. Here, the tube length is assumed to be constant, and the output power is saturated by continuously increasing the input power. In CST, the grid number of the SDSG-TWT is set to 18,000,000 and the time required by the PC (2.9 GHz CPU and Tesla k20c accelerator card) is 25 h for a 12 ns simulation of a single input signal. The results are displayed in Figure 11, Figure 12, Figure 13, Figure 14, Figure 15, Figure 16, Figure 17 and Figure 18.

Figure 11 shows the variation of signal amplitude over time at 340 GHz for TWTs using the SDSG-SWS. The results show that the SDSG-TWT reaches a stable amplification state after 0.8 ns and remains without oscillation. The SDSG-TWT achieves an output voltage of 8.1 V (corresponding power of 32.8 W) at an input voltage of 0.35 V (corresponding power of 0.06 W).

Figure 12 presents the energy distribution of electrons in the phase space along the longitudinal direction when the signal remains at stable amplification for a long time. The results show that there are more decelerating electrons than accelerating electrons. Most of the electronic energy is converted into the energy of the electromagnetic wave. It can be observed that the electromagnetic wave signal is amplified.

Figure 13 is a full cycle electric field diagram. The results show that, as the longitudinal distance increases, the electric field intensity within the SWS also increases, which indirectly confirms that the SDSG-TWT can effectively amplify the input signal.

Figure 14 shows the longitudinal and transverse cross-sectional views of the electron beam (The cross-sectional view shows the connection between the SWS and the output coupler). The longitudinal cross-sectional view shows that electric field energy increases with increasing longitudinal distance. At the same time, near the end of the SWS circuit, electronic modulation reaches saturation. This result is consistent with the previous phase space diagram. The cross-sectional view shows that the electrons are not near the red line around them (The red line indicates the size of the electron beam channel). This result indicates that the modulated electrons were not intercepted by the metal wall.

Figure 15 shows the spectrum of the output signal. The Fourier transform of the output signal shows that that, with the exception of 340 GHz, the signals’ amplitudes at other frequencies are extremely low, to the extent that they can be ignored. It indicates that the SDSG-TWT can effectively amplify the fundamental signal of 340 GHz without the oscillation starting of other signals.

Figure 16, Figure 17 and Figure 18 show the performance comparison between the SDSG-TWT and the SW-TWT in terms of their saturated output power, gain, and electron efficiency. The results shown in Figure 16, Figure 17 and Figure 18 indicate that the saturated output powers of the SDSG-TWT and SW-TWT are 32.8 W and 23.1 W; the 3 dB bandwidths are 316 GHz–405 GHz and 315 GHz–370 GHz; the maximum gains are 1.19 dB/mm and 0.61 dB/mm; and the maximum electron efficiencies are 2.84% and 1.80%, respectively. According to these results, it can be calculated that, compared with the SW-TWT, the SDSG-TWT demonstrates a 41% improvement in saturated output power, a 61.8% improvement in 3 dB bandwidth, an 83% improvement in gain, and a 63.3% improvement in electron efficiency under the same operating conditions.

Table 2 shows a comparison of the performance between the proposed SDSG and three reported improved SWs. Compared with the new SW-SWS [21], the SDSG-SWS demonstrates significant advantages in operating voltage, operating current, gain, output power, and electronic efficiency, due to the MBSC-SWG-SWS [22] being a multi electron beam channel structure. Therefore, compared to the MBSC-SWG-SWS, the SDSG-SWS only has certain advantages in terms of gain. However, for 340 GHz TWT, it is very difficult to design the electron optics system of a multi-beam TWT. Therefore, the structure presented in this article is more applicable and practical. Compared with the modified SW-SWS [13], the SDSG-TWT has excellent performance in all aspects.

In summary, the SDSG-TWT demonstrates significant improvements in saturated output power, gain, and electron efficiency compared with the SW-TWT. PIC simulation results further validate the performance advantages of the SDSG-SWS over the SW-SWS.

## 4. Conclusions

A new SWS, called staggered double-segmented grating (SDSG), which is a combination of the SW-SWS and SDG-SWS, is investigated. Research has shown that it has the following characteristics: wide operating band, high interaction impedance, low loss, and ease of fabrication. Compared with the SW-TWT, the SDSG-TWT can produce higher output power, greater gain, and electron efficiency under the same operating conditions. Therefore, the SDSG-SWS can be regarded as a very promising submillimeter TWT slow-wave circuit.

## Figures and Tables

**Figure 1 sensors-23-04762-f001:**
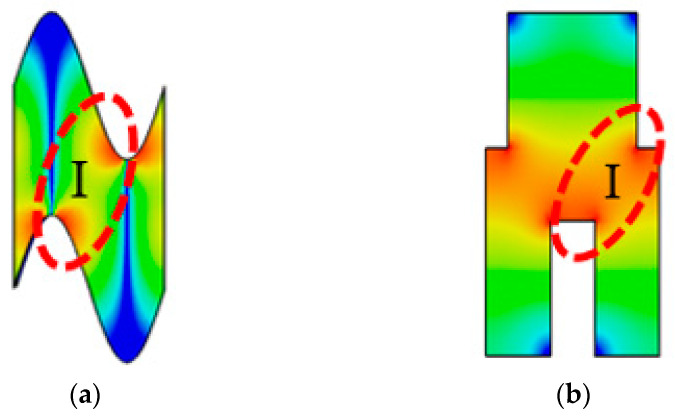
Distribution of longitudinal electric field $E_{Z}$ for (**a**) SW-SWS and (**b**) SDG-SWS.

**Figure 2 sensors-23-04762-f002:**
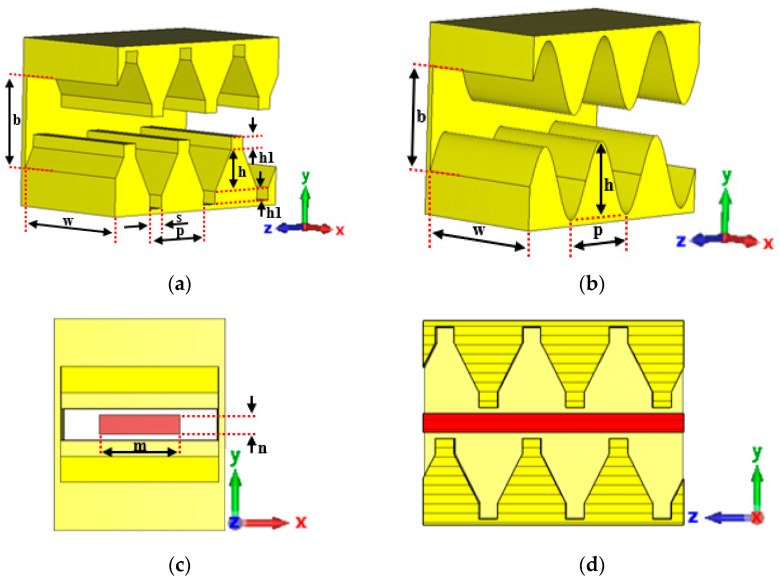
Solid models without side wall at *X*-max of (**a**) SDSG-SWS and (**b**) SW-SWS; (**c**) left-view and (**d**) sectional view in the *y-o-z* plane of SDSG-SWS with the sheet beam.

**Figure 3 sensors-23-04762-f003:**
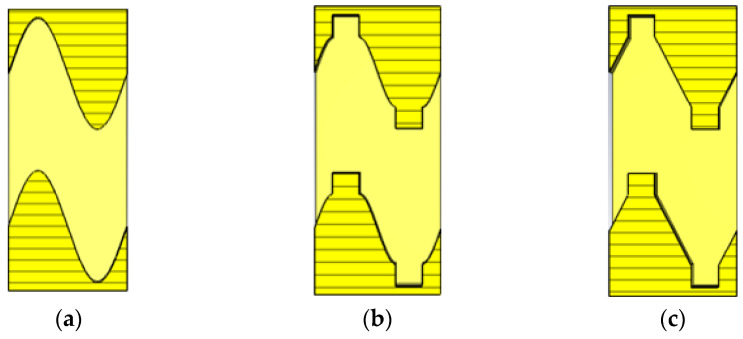
Sectional views in the *y-o-z* plane of (**a**) SW-SWS, (**b**) transition structure, and (**c**) SDSG-SWS.

**Figure 4 sensors-23-04762-f004:**
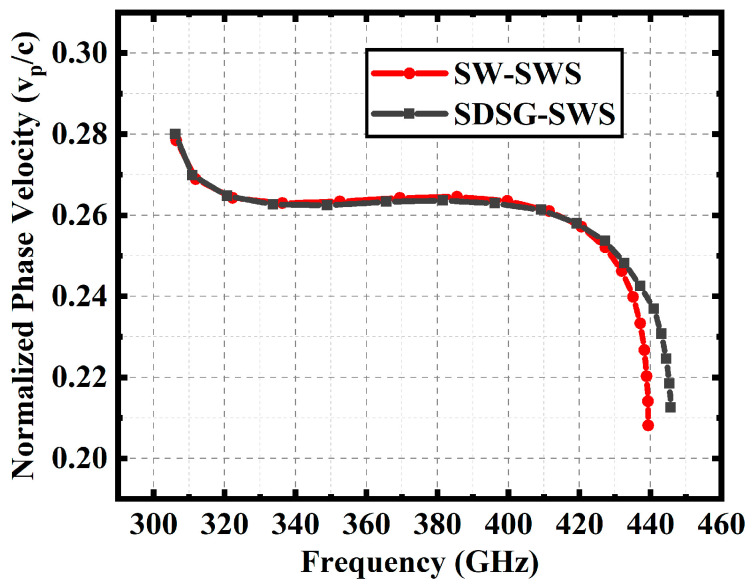
Dispersion curves for SDSG-SWS and SW-SWS.

**Figure 5 sensors-23-04762-f005:**
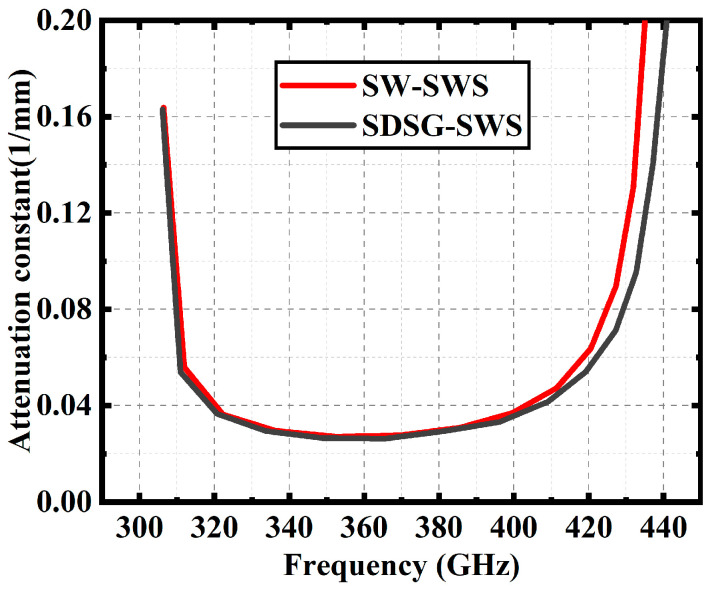
Attenuation constant curves of SDSG-SWS and SW-SWS.

**Figure 6 sensors-23-04762-f006:**
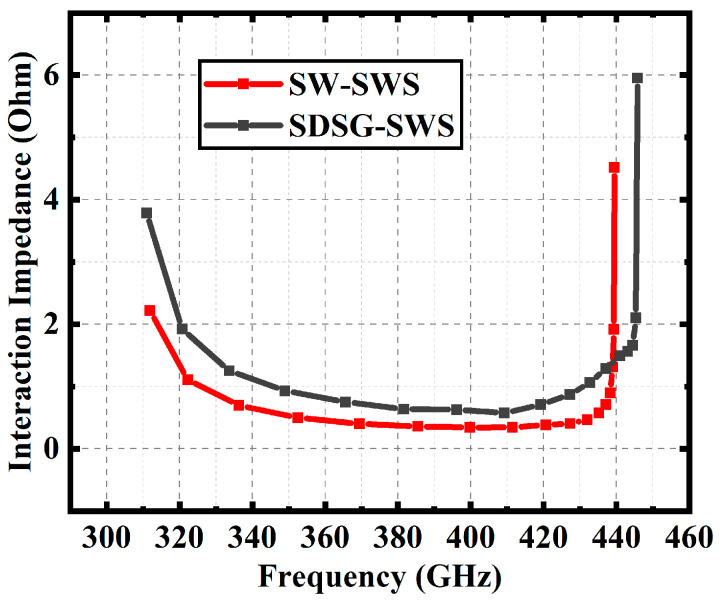
Average interaction impedance curves of SDSG-SWS and SW-SWS.

**Figure 7 sensors-23-04762-f007:**
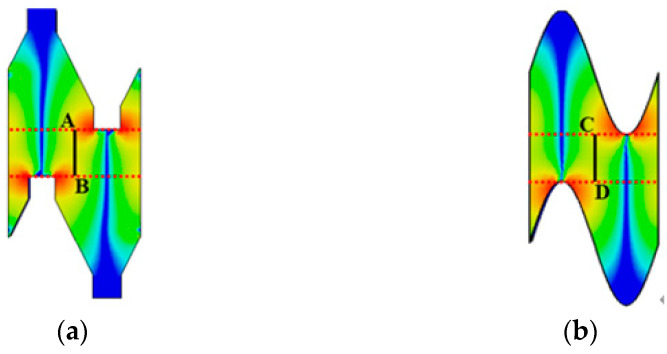
Distribution of $E_{Z}$ in the *y-o-z* plane of (**a**) SDSG-SWS, (**b**) SW-SWS at 340 GHz.

**Figure 8 sensors-23-04762-f008:**
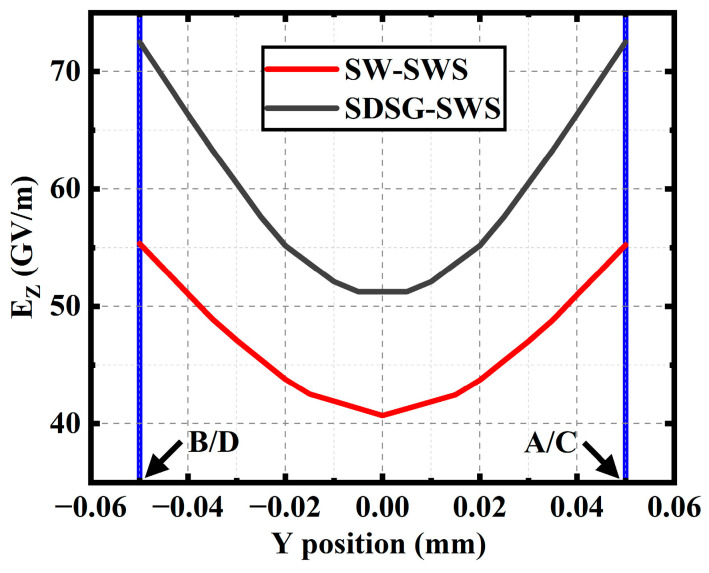
Variation of $E_{Z}$ in the *y-o-z* plane along the *y*-direction (A–B and C–D).

**Figure 9 sensors-23-04762-f009:**
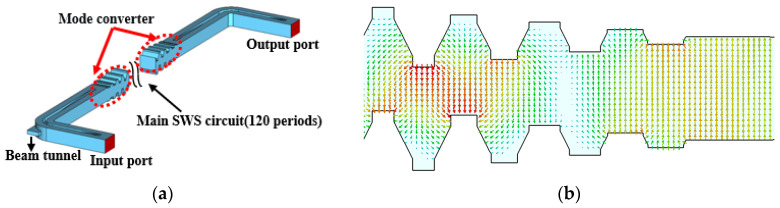
(**a**) Vacuum model of SDSG slow-wave circuit with couplers. (**b**) Cross-sectional view of the coupler electric field in the *y-o-z* plane.

**Figure 10 sensors-23-04762-f010:**
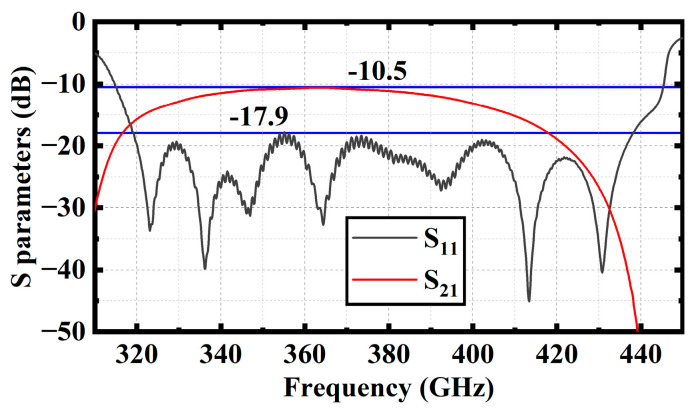
Transmission characteristics of SDSG slow-wave circuit with couplers.

**Figure 11 sensors-23-04762-f011:**
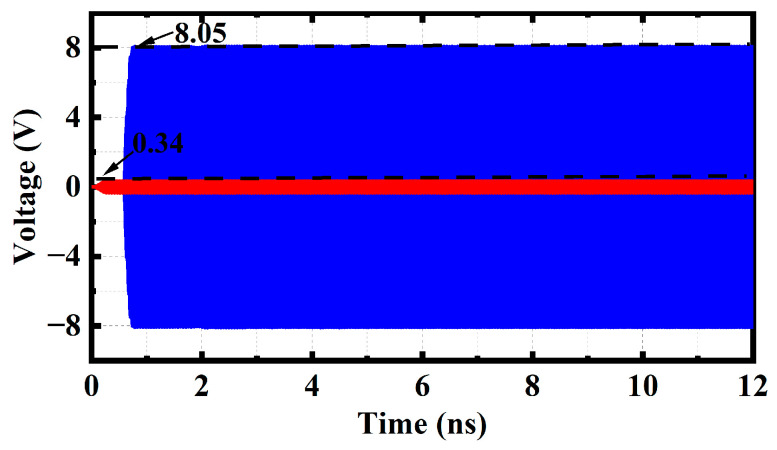
The signal amplitude versus time at 340 GHz.

**Figure 12 sensors-23-04762-f012:**
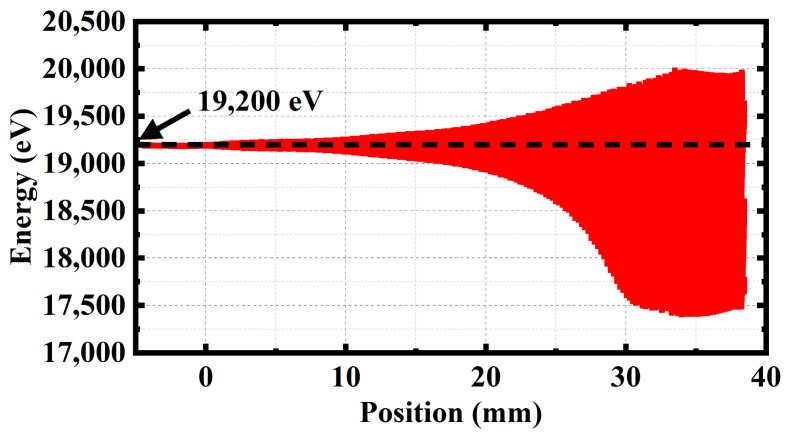
Phase momentum plot of the electrons.

**Figure 13 sensors-23-04762-f013:**
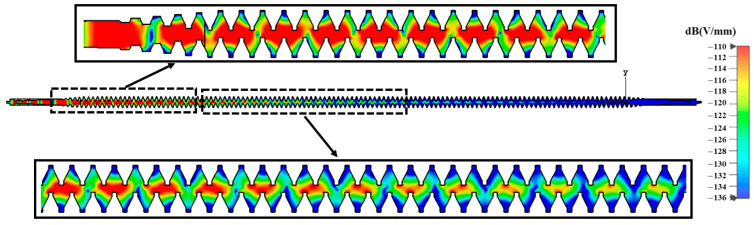
Electric field cross-section (*y*-direction).

**Figure 14 sensors-23-04762-f014:**
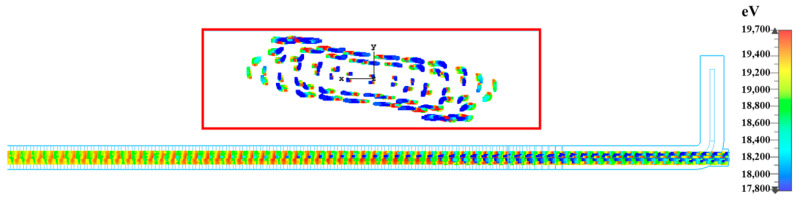
Transverse and longitudinal cross-sectional view of the electron trajectory.

**Figure 15 sensors-23-04762-f015:**
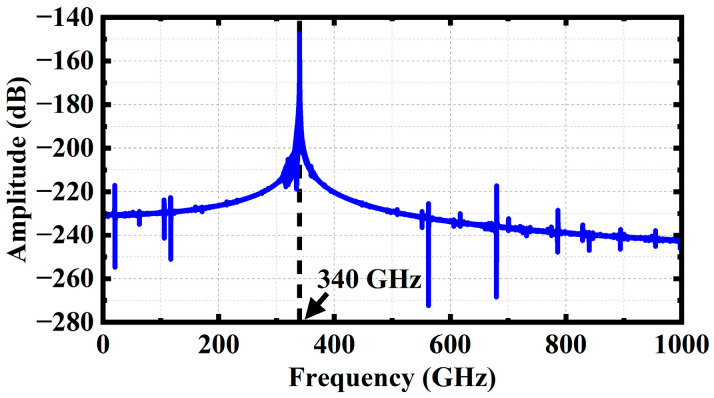
Frequency spectrum of output signal.

**Figure 16 sensors-23-04762-f016:**
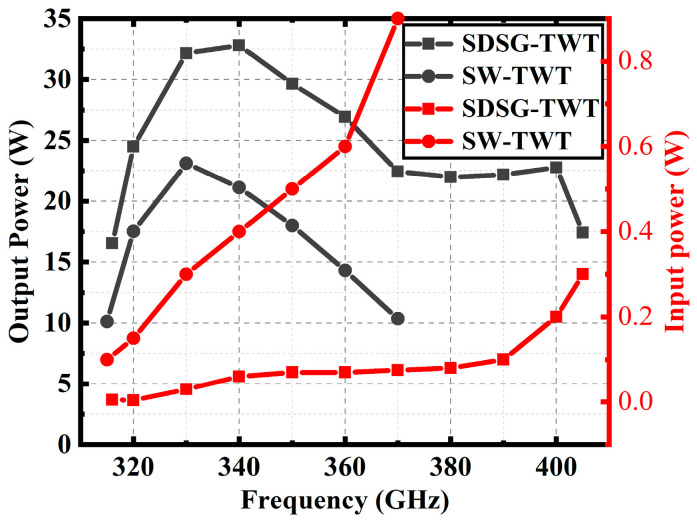
Output–input power versus frequency.

**Figure 17 sensors-23-04762-f017:**
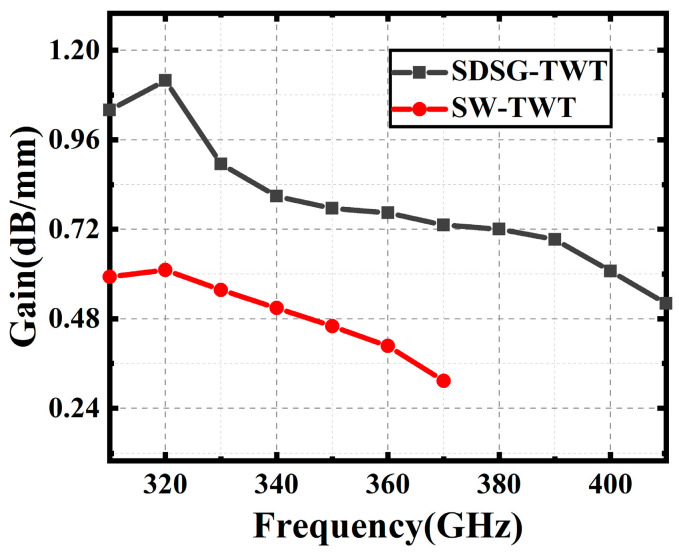
Gain versus frequency.

**Figure 18 sensors-23-04762-f018:**
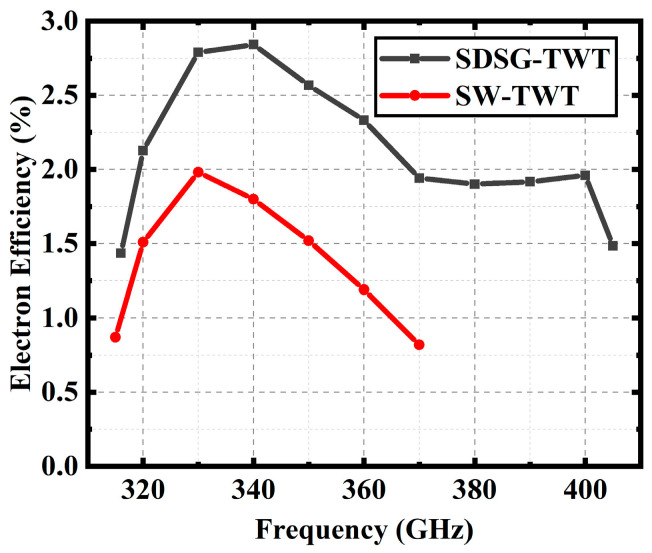
Electron efficiency versus frequency.

**Table 1 sensors-23-04762-t001:** Optimal parameters of SDSG-SWS and SW-SWS.

Parameter	Value (mm)	
	SDSG-SWS	SW-SWS
*p*	0.282	0.282
*b*	0.36	0.37
*w*	0.49	0.49
*h*	0.16	0.27
*h1*	0.05	/
*s*	0.06	/
*m*	0.25	0.25
*n*	0.1	0.1

**Table 2 sensors-23-04762-t002:** Comparison of SDSG-SWS with the published SWS at 340 GHz.

Structure	V (kV)	I (mA)	Outpower (W)	Gain (dB)	η (%)
SDSG	19.2	60	32.8	37.87	2.84
New SW [21]	12.65	30	10	20	2.63
MBSC-SWG [22]	21.3	54	51	24	4.43
Modified SW [13]	9.9	40	10	27	2.52

## Data Availability

Data sharing is not applicable.

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
