# Peer review of "A Novel Staggered Double-Segmented Grating Slow-Wave Structure for 340 GHz Traveling-Wave Tube"

_sensors, 2023, doi:10.3390/s23104762_

Round 1
Reviewer 1 Report
A novel Staggered Double Segmented-Grating Slow Wave Structure for THz TWT is proposed in the manuscript. The SDSG improves the saturated power, gain, and efficiency compared with the SW-TWT. The manuscript is well organized and presented. And some comments are listed below.
1) Line 127, it indicates the simulation is carried by ANSOFT and CST. But most of the results are more likely by the CST. Could you please indicate more clearly?
2) Line 159, figure 8 gives a magnitude of 1010 V/m, which is relatively high and induces breakdown. Can you explain the setup in the simulation?
3) Line 162, The detailed structure and electric field distribution for couplers should be included.
4) Please gives the electric field distribution and electron trajectory for PIC simulation.
5) Please compare the performance in this paper with the published articles in 340 GHz TWT.
Author Response
First of all, I would like to send my best wishes to you for your objective and fair comments on my research paper. According to your comments, I examined and modified the paper carefully.
Please see the attachment

Reviewer 2 Report
1. It would be more helpful and would also guaranteed the self-autonomous character of the paper, if the authors added some more theoretical aspects regarding their concept. In this manner, the paper will be more helpful for the interested reader.
2. Can the authors provide some evidence regarding their numerical simulations? For example, what is the required RAM or CPU time?
3. Have the authors attempted to proceed to some experimental results? In such a case a description of the experimental setup would be indeed helpful.
4. How the proposed structure compares to existing ones from the literature apart from those appearing in the paper?
The language of the manuscript is adequate.
Author Response
First of all, I would like to send my best wishes to you for your objective and fair comments on my research paper. According to your comments, I examined and modified the paper carefully.
Please see the attachment.

Round 2
Reviewer 2 Report
The revised version of the paper has been improved. Thus, I think that the paper can be accepted for publication.
The language is acceptable and can be understood.